# The Value of Low Prostate Imaging—Reporting and Data System (PI-RADS) Scores in Preventing Unnecessary Prostate Biopsies

**DOI:** 10.3390/medicina57050413

**Published:** 2021-04-24

**Authors:** Dong-Soo Kim, Sung-Kyoung Moon, Joo-Won Lim, Seung-Hyun Jeon, Sang-Hyub Lee

**Affiliations:** 1Department of Urology, Kyung Hee University School of Medicine, Seoul 02453, Korea; dskurology@gmail.com (D.-S.K.); juro@khu.ac.kr (S.-H.J.); 2Department of Radiology, Kyung Hee University School of Medicine, Seoul 02453, Korea; aquamsk@naver.com (S.-K.M.); limjoone@naver.com (J.-W.L.)

**Keywords:** prostate cancer, multiparametric magnetic resonance imaging, image-guided biopsy, prostate specific antigen, cancer screening

## Abstract

*Background and Objectives*: Magnetic resonance imaging (MRI) and the Prostate Imaging-Reporting and Data System (PI-RADS) have become essential tools for prostate cancer evaluation. We evaluated the ability of PI-RADS scores in identifying significant prostate cancer, which would help avoid unnecessary prostate biopsies. *Materials and Methods*: Patients with prostate-specific antigen (PSA) levels ≤ 20 ng/mL, who underwent prostate MRI for evaluation from January 2018 to November 2019, were analyzed. Among them, 105 patients who received transrectal ultrasonography (TRUS)-guided biopsy were included. PSA, PI-RADS scores (low 1–2, high 3–5), biopsy results, and Gleason scores (GS) were evaluated. Biopsies with GS higher than 3 + 4 were considered as significant cancers and biopsies with no cancer or Gleason 3 + 3 were considered insignificant or no cancers. *Results*: Among the 105 patients, 45 patients had low PI-RADS and 60 had high PI-RADS scores. There were no patients with significant prostate cancer in the low PI-RADS groups. For the high PI-RADS group, 28 (46.7%) patients had significant cancer and 32 (53.3%) had insignificant or no cancer. The sensitivity and specificity of high PI-RADS to detect significant cancer was 100% and 58.4%, respectively. Positive predictive value was 46.7% and negative predictive value was 100%. *Conclusions*: Low PI-RADS scores on MRI did not show significant prostate cancer and surveillance should be considered in selected cases to prevent unnecessary invasive procedures and overdiagnosis.

## 1. Introduction

Screening for prostate cancer is a controversial topic [1]. Appropriately timed diagnosis is essential for optimal treatment and management of any disease. However, prostate screening can result in overdiagnosis and consequent overtreatment, which increases the incidence of post-surgical or post-radiotherapy complications, such as impotence and incontinence [2,3]. Standard transrectal ultrasonography (TRUS)-guided prostate biopsy can also cause severe post-procedure complications which may prolong hospitalization and treatment duration [4]. Therefore, careful patient evaluation and counseling is important before making a decision to subject the patient to prostate cancer screening.

Magnetic resonance imaging (MRI) is an essential tool for prostate cancer evaluation. Traditionally, MRI was used for staging evaluation in biopsy-proven prostate cancer [5]. Recent advances in technology have allowed MRI to become an even more powerful tool. Multiparametric MRI (mpMRI) adds functional imaging forms such as dynamic contrast-enhanced (DCE) MRI, diffusion-weighted imaging (DWI), and calculated apparent diffusion co-efficient (ADC) maps to standard anatomical T1- and T2-weighted imaging [6]. Efforts to standardize and facilitate prostate MRI interpretation have also rapidly progressed and, currently, the Prostate Imaging—Reporting and Data System (PI-RADS) has been updated to version 2.1 (v2.1) [7].

PI-RADS aims to help localize and characterize prostate cancer in pre-treated patients and select patients for biopsy and future management. This will help distinguish between insignificant and significant cancer and aid in deciding the need for biopsy or re-biopsy [8]. However, validation is still needed to prove the accuracy and inter-interpreter consistency of PI-RADS [9].

In this study, we aimed to contribute to the ongoing accumulation of data on PI-RADS validation. We matched the PI-RADs scores of suspicious prostate MRI lesions with concurrent prostate biopsy results. The ability of PI-RADS to predict clinically significant prostate cancer was evaluated. Thus, we also examined the possibility of using PI-RADS results to prevent unnecessary prostate biopsies.

## 2. Materials and Methods

The medical records of biopsy naïve patients with prostate-specific antigen (PSA) levels ≤ 20 ng/mL who underwent prostate MRI for prostate cancer evaluation from January 2018 to November 2019 at Kyung Hee University Medical Center (KHMC) were retrospectively analyzed. The initial number of patients selected for this study was 229. Among them, 105 patients who received TRUS-guided biopsy were included into the study. Pre-biopsy PSA, MRI PI-RADS scores, post-biopsy cancer diagnosis, and Gleason scores (GS) were evaluated. Patient evaluation and decisions regarding prostate biopsy were made by 5 urologists at KHMC. A comprehensive approach that included consideration of changes in PSA values, transrectal ultrasonography findings, MRI findings and their PI-RADS scores were used when deciding on the biopsy. The significance of the findings, the pros and cons of prostate biopsy, and the implications of the biopsy results were carefully explained to the patients, and biopsy was conducted after consent.

The Philips Achieva 3.0T MRI scanner (Philips Healthcare, Andover, MA) was used for evaluation. Sequences used included T1-weighted images (axial), T2-weighted images (axial, coronal, sagittal) and DWI/ADC maps (axial, upper abdomen, b-values: 0, 800, 1400, 2000). For contrast enhancement, DCE axial plane images and fat suppression T1 axial images were taken. Prostate MRI findings were reported by 2 radiologists specializing in urologic radiology who had access to the patient’s medical records. The latest available update of PI-RADS was used. The PI-RADS scores of the transitional zone and peripheral zone were both initially reported, but for the purpose of this study, only the higher score was considered for analysis.

For statistical evaluation, patients were divided into two groups according to their PI-RADS scores. The low PI-RADS groups was defined as patients with PI-RADS scores of 1 or 2, and the high PI-RADS groups were assigned to those with PI-RADS scores from 3 to 5. The PI-RADS scores of the patients were matched with prostate biopsy results. Significant prostate cancer was defined as cancer with GS 7 or higher, while insignificant or no cancer was considered in cases of negative biopsies or cancer with GS of 3+3 [10].

All prostate biopsies were conducted transrectally by the same 2 radiologists who reported the MRIs using ultrasound guidance. Prostate biopsy was usually performed with a 12-core template and when needed, additional cores were taken to evaluate suspicious lesions. The 12-core template that was used for a standard biopsy was as follows: 1—right base lateral, 2—right middle lateral, 3—right apex lateral, 4—left base lateral, 5—left middle lateral, 6—left apex lateral, 7—right base medial, 8—right middle medial, 9—right apex medial, 10—left base medial, 11—left middle medial, and 12—left apex medial. Patients were given antibiotics and an enema was conducted before biopsy and they were discharged after confirmation of no severe post-biopsy complications.

Statistical analysis was performed using SPSS software (Statistical Package for the Social Sciences, V. 16.0; SPSS Inc., Chicago, IL, USA). Means and standard deviations of baseline patient characteristics and biopsy results were analyzed, and statistical significances was evaluated calculated using t-tests and Chi-square tables. The sensitivity, specificity, positive predictive value (PPV), and negative predictive value (NPV) of PI-RADS to detect insignificant or no prostate cancer were also analyzed.

This study was reviewed by our institutional review board and approved due to its retrospective nature and its application of standard practice.

## 3. Results

### 3.1. Baseline Patient Characteristics

A total of 105 patients who underwent prostate MRI and prostate biopsy were included in this study. Mean age was 68.28 ± 8.579, and mean PSA was 8.01 ± 4.2771 ng/mL. The mean number of cores taken were 13.03 ± 1.397, and the mean number of positive cores were 2.24 ± 2.976. Seven (6.7%) patients had PI-RADS 1, 38 (36.2%) had PI-RADS 2, 13 (12.4%) had PI-RADS 3, 20 (19%) had PI-RADS 4, and 27 (25.7%) had PI-RADS 5 scores.

Fifty-one (48.6%) patients had negative biopsy results and 54 (51.4%) were positive for prostate cancer. Among the 54 cancer patients, 26 (48.1%) had GS 3+3, 7 (13%) had GS 3+4, 8 (14.8%) had GS 4+3, 10 (18.5%) had GS 4+4, 1 (1.9%) had GS 4+5, and 2 (3.7%) had GS 5+4. When divided into clinical significance, 77 (73.3%) patients had insignificant or no cancer and 28 (26.7%) patients had significant cancer.

### 3.2. Comparison between Low and High PI-RADS Score Groups

Patients were divided into two groups according to their highest reported PI-RADS score. Forty-five patients were placed in the low PI-RADS score group and 60 in the high PI-RADS score group. The characteristics of each group were analyzed. Patients in the high PI-RADS score group were older (low vs. high, 66.20 ± 9.576 vs. 69.77 ± 7.576, *p* = 0.044), had more cores taken during biopsy (12.50 ± 0.952 vs. 13.43 ± 1.544, *p* < 0.001), and showed higher number of positive cores (0.30 ± 0.765 vs. 3.70 ± 3.180 *p* < 0.001). There was no difference in PSA (7.71 ± 4.097 vs. 8.27 ± 4.441, *p* = 0.059) between the two groups.

Eight patients in the low PI-RADS score group and 46 in the high PI-RADS score group were diagnosed with prostate cancer. All cancer patients in the low PI-RADS score group had a GS of 3+3. 18 (39.1%) of the cancer patients in the high PI-RADS score group had a GS of 3+3 and 28 (60.9%) had GS 3+4 and higher. The results of the analysis of the two groups are summarized in Table 1.

### 3.3. Detection of Clinically Significant Prostate Cancer in Low and High PI-RADS Groups

Among the total of 105 patients, 28 (26.7%) had clinically significant prostate cancer and 77 (73.3%) had insignificant or no cancer. We stratified significant and insignificant or no cancer cases according to PI-RADS scores. There were no patients with significant prostate cancer in the low PI-RADS groups. All patients in the low PI-RADS group were shown to have insignificant or no cancer (45, 100%). Among them, 8 (17.8%) patients had Gleason score 3+3 prostate cancer and 37 (82.2%) had no cancer on biopsy. For the high PI-RADS group, 28 (46.7%) had significant cancer and 32 (53.3%) had insignificant or no cancer. Among the high PI-RADS insignificant cancer patients, 18 (30%) had Gleason score 3+3 cancer, and 14 (23.3%) had no prostate cancer. The sensitivity and specificity of high PI-RADS to detect significant cancer was 100% and 58.4%, respectively. Positive predictive value was 46.7% and negative predictive value was 100%. The results are presented in Table 2.

We also analyzed the proportion of significant cancer patients and insignificant or no cancer patients for each PI-RADS score in the high PI-RADS group. Among the 60 patients of the high PI-RADS group, 13 (21.7%) were PI-RADS 3, 20 (33.3%) were PI-RADS 4, and 27 (45%) were PI-RADS 5. The incidence of significant cancer vs. insignificant or no cancer for each score were PI-RADS 3, 3 (23.1%) vs. 10 (76.9%), PI-RADS 4, 10 (50%) vs. 10 (50%), and PI-RADS 5, 15 (55.6%) vs. 12 (44.4%).

## 4. Discussion

Since 2012, PI-RADS has served as a platform to unify the interpretation and reporting of prostate MRIs [6]. The well-known 5-point scale provides a concise and direct description of prostate lesions and allows easier interdisciplinary discussion. Several studies have already proven the ability of PI-RADS in diagnosing significant cancer in high scores and excellent negative predictive value for significant cancers in low scores [9]. Updated to its second version in 2015, and version 2.1 in 2019, PI-RADS is a “living document” that strives to optimize prostate MRI acquisition and reduce interobserver variability [11].

Our study adds supporting evidence of the ability of PI-RADS to exclude insignificant prostate cancer. None of the patients with low PI-RADS scores were diagnosed with significant cancer (NPV 100%). The well-known Prostate MRI Imaging study or PROMIS study has shown the negative predictive power of mpMRI [12]. Futterer et al. reviewed the available literature on the ability of mpMRI to detect clinically significant prostate cancer and showed that the NPV and PPV for the detection of significant cancer was 63% to 98% and 34% to 68%, respectively [13]. The high NPV of low PI-RADS is important because it signifies that clinically low risk patients may be excluded or delayed from prostate biopsy. The 2019 European Association of Urology (EAU) guidelines [1] and the United Kingdom National Institute for Health and Care Excellence (NICE) [14] recommends no biopsy after low-likelihood MRI without high clinical risk. Further, the PI-RADS Steering Committee stated that in biopsy-naïve men, a no-immediate-biopsy approach should be considered in patients with low-likelihood MRI after careful discussion with the patient about the pros and cons of biopsy [15].

The PPV of high PI-RADS for significant prostate cancer in this study was 46.67%. Compared with NPV, the PPV is relatively low but still within the reported interval (34%–68%) mentioned above [13]. The low PPV can be attributed to PI-RADS 3 lesions and to the fact that most of the biopsies included in this study were systemic TRUS-guided biopsies. Analysis of PI-RADS 3 lesions in our study showed that 23.1% had clinically significant prostate cancer, which was higher than previously reported studies such as those by Santhianathen et al [16]. (8.9%) and Mehralivand et al. [17] (12%). Despite low positive predictive rates, the EAU 2019 and NCCN 2019 guidelines encourage combined systematic and targeted biopsies in biopsy naïve men after intermediate-likelihood MRI findings. Furthermore, the PI-RADS Steering Committee [15] also recommends PI-RADS 3 as a positive scan requiring peer-review and biopsy. MRI-guided biopsies [18] have been reported to be able to detect clinically significant prostate cancer as accurately as standard TRUS guided biopsies with lower insignificant case detection. An MRI lesion-guided biopsy approach could have led to a higher detection rate for this group. In addition, as the PRECISION (Prostate Evaluation for Clinically Important Disease: Sampling Using Image Guidance or Not?) study shows, review by experts and experience after time can reduce PI-RADS 3 percentage and future refinements will aid in increasing accuracy [19].

Our paper has several limitations which should be considered in future studies. First, this is a retrospective analysis of the medical reports of a limited number of patients who had undergone prostate biopsy. A prospective study with a larger patient pool will be needed to confirm the accuracy of our data and the predictability of the PI-RADS system. Second, our study did not consider interobserver differences in PI-RADS reporting. The reported mpMRIs were examined by two radiologists specializing in urologic radiology, but we did not compare variations in observation between the two, so no data on interobserver consistency were acquired. Moreover, differences due to the update of the PI-RADS version were not considered during analysis.

Despite these limitations, our data sufficiently represent the ability of low-grade PI-RADS to rule out insignificant or no cancer in biopsy naïve patients. The negative predictive value of PI-RADS is very high and, thus, should be considered a determining factor when discussing the need for a prostate biopsy. However, PI-RADS is not an all-powerful method of prediction. The low specificity of high PI-RADS scores or our results shows that while PI-RADS is a useful tool for in evaluation, it should not be considered as a definitive method for diagnosing prostate cancer, and a biopsy is still necessary to confirm the cancer’s existence.

## 5. Conclusions

Low PI-RADS scores seen on mpMRI have a high NPV for clinically significant prostate cancer. Patients presenting with low PI-RADS on mpMRI should be counseled on its significance and a delayed prostate biopsy can be offered to prevent overtreatment, overdiagnosis, and biopsy-related complications.

## Figures and Tables

**Table 1 medicina-57-00413-t001:** Representation of baseline patient information and distribution of Gleason score in patients diagnosed with prostate cancer.

Variables ^a^	Low PI-RADS ^b^ (*n* = 45)	High PI-RADS ^c^ (*n* = 60)	*p*-Value
Age (years)	66.20 ± 9.576	69.77 ± 7.576	0.044
PSA (ng/mL)	7.71 ± 4.097	8.27 ± 4.441	0.509
Total cores	12.50 ± 0.952	13.43 ± 1.544	0.000
Positive cores	0.30 ± 0.765	3.70 ± 3.180	0.000
Cancer			0.000
No	37 (82.2%)	14 (23.3%)	
Yes	8 (17.8%)	46 (76.7%)	
Gleason Scores			0.001
3+3	8 (100%)	18 (39.1%)	
3+4 ≤	0 (0%)	28 (60.9%)	
Exact Gleason Scores			
3+3	8 (100%)	18 (39.1%)	
3+4	0 (0%)	7 (15.2%)	
4+3	0 (0%)	8 (17.4%)	
4+4	0 (0%)	10 (21.7%)	
4+5	0 (0%)	1 (2.2%)	
5+4	0 (0%)	2 (4.3%)	

^a^ Data are presented as mean ± standard deviation or number (percentage); ^b^ Low PI-RADS scores were defined as 1–2; ^c^ High PI-RADS score were defined as 3–5; Abbreviations: PI-RADS, Prostate Imaging—Reporting and Data System; PSA, prostate-specific antigen.

**Table 2 medicina-57-00413-t002:** Sensitivity, specificity, positive predictive value, and negative predictive value of high versus low PI-RADS to detect significant prostate cancer.

	Cancer		No Cancer	Total
	GS 3+4	GS 3+3		
High PI-RADS	28 (46.7%)	18 (30%)	14 (23.3%)	60
Low PI-RADS	0 (0%)	8 (17.8%)	37 (82.2%)	45
	Significant cancer	Insignificant or no cancer		
High PI-RADS	28	32		PPV 46.7%
Low PI-RADS	0	45		NPV 100%
	Sensitivity 100%	Specificity 58.4%		

Abbreviations: GS, Gleason score; PI-RADS, Prostate Imaging—Reporting and Data System; PPV, Positive predictive value; NPV, Negative predictive value.

## Data Availability

The data that support the findings of this study are available on request from the corresponding author. The data are not publicly available due to privacy or ethical restrictions.

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
