# Peer review of "The Value of Low Prostate Imaging—Reporting and Data System (PI-RADS) Scores in Preventing Unnecessary Prostate Biopsies"

_medicina, 2021, doi:10.3390/medicina57050413_

Round 1
Reviewer 1 Report
Well written paper. Real world evidence that shows a local institutions experience with mpMRI for prostate cancer analysis. Retrospective study. Does not describe if quality of MRI was considered though 2 radiologists review is appropriate. Relatively small study. Data supports literature and current consensus on use and value of mpMRI. Good to have real world data confirming controlled trial data and also good for institutions to determine their own sensitivity and specificity of diagnostic tests in their own hands. Would be good to clearly state in the paper definition of low PIRADS vs high PIRADS - I found it in a table caption and should be clearly stated in the methods.
Author Response
<Reviewer 1>
Well written paper. Real world evidence that shows a local institutions experience with mpMRI for prostate cancer analysis. Retrospective study. Does not describe if quality of MRI was considered though 2 radiologists review is appropriate. Relatively small study. Data supports literature and current consensus on use and value of mpMRI. Good to have real world data confirming controlled trial data and also good for institutions to determine their own sensitivity and specificity of diagnostic tests in their own hands. Would be good to clearly state in the paper definition of low PIRADS vs high PIRADS - I found it in a table caption and should be clearly stated in the methods.
<Answer to Reviewer 1>
Thank you for your kind and insightful review. Theses are our responses to your questions and comments.
On page 2 line 71, we described the parameters we used to classify the two groups of interest. PI-RADS score 1-2 were assigned as low PIRADS score group and 3-5 as high score group. We revised the manuscript so that it could better convey the definition of the two groups.
Reviewer 2 Report
Comments to the authors
1) General comments
This paper features the significance of PI-RADS score of prostate MRI for discriminating insignificant prostate cancer. The authors have examined 100 cases and drawn a reasonable conclusion. However, I’m afraid that there is a severe problem with this paper.
- p.2, l.74
“while insignificant cancer was considered in cases of negative biopsies or cancer with GS of 3+3.”
Basically, a negative biopsy is not an insignificant cancer. A negative biopsy means that there is no cancer, but it is not an insignificant cancer. This classification is not allowed.
- p.2 Material and Methods
What is the inclusion criteria of prostate biopsy? The patients with low PI-RADS (<=2) may not be candidates for prostate biopsy in the first place. If you didn’t consider MRI findings for selecting for candidates for prostate biopsy, what did you choose base on? You should clarify.
- p.2 l.78
You stated that prostate biopsy was usually done with a 12-core template biopsy. What kind of template biopsy did you adapt? You should clarify.
- p.2
What are the conditions of MRI imaging? What is the b-value of DWI? You should clarify.
2) Specific comments for revision
- p.1 Abstract
In your abstract, there is no definition of a low PI-RADS score. You should mention in the abstract. For example, the low PI-RADS (<=2).
Author Response
<Answers to reviewer 2>
Thank you for reviewing our manuscript and giving us your detailed comments and in-depth inquires concerning it. We have reviewed them and hope our responses are sufficient in answering them.
1) General comments
1. p.2, l.74 Insignificant cancer.
“while insignificant cancer was considered in cases of negative biopsies or cancer with GS of 3+3.” Basically, a negative biopsy is not an insignificant cancer. A negative biopsy means that there is no cancer, but it is not an insignificant cancer. This classification is not allowed.
As you have commented, we agree that a negative biopsy signifies that there is no cancer and should not be considered as an insignificant cancer. The goal of this paper was to see if PI-RADS scores could be helpful during pre-biopsy evaluation in discovering patients who would benefit from undergoing prostate biopsy and prevent unnecessary invasive procedures. The “insignificant cancer” group was meant to represent cases where prostate biopsy could be delayed or should not be considered mandatory for evaluation. The manuscript was revised to better describe this group comprehensively and we specifically mentioned negative biopsies (patients with no cancer) and insignificant cancer biopsies as two different entities.
2. p.2 Material and Methods, Inclusion criteria of prostate biopsy.
What is the inclusion criteria of prostate biopsy? The patients with low PI-RADS (<=2) may not be candidates for prostate biopsy in the first place. If you didn’t consider MRI findings for selecting for candidates for prostate biopsy, what did you choose base on? You should clarify.
Thank you. 5 urologists evaluated all patients and after initial evaluation, the results were discussed with each patient. A comprehensive approach that included consideration of changes in PSA values, transrectal ultrasonography findings, MRI findings and their PI-RADS scores were used when deciding on the biopsy. The significance of the findings, the pros and cons of prostate biopsy, and the implications of the biopsy results were carefully explained to the patients, and biopsy was done after consent. We have this explanation to the manuscript.
3. p.2 l.78
You stated that prostate biopsy was usually done with a 12-core template biopsy. What kind of template biopsy did you adapt? You should clarify.
Thank you for this comment. The 12-core template we use for a standard biopsy was as follows; 1: right base lateral, 2: right middle lateral, 3: right apex lateral, 4: left base lateral, 5: left middle lateral, 6: left apex lateral, 7: right base medial, 8: right middle medial, 9: right apex medial, 10: left base medial, 11: left middle medial, 12: left apex medial. We have added this to the manuscript to clarify the biopsy template schematics.
4. p.2 Conditions of MRI imaging, b-value of DWI
What are the conditions of MRI imaging? What is the b-value of DWI? You should clarify.
Thank you for letting us clarify. The Philips Achieva 3.0T MRI scanner (Philips Healthcare, Andover, MA) was used for evaluation. Sequences used included T1 weighted images (axial), T2 weighted images (axial, coronal, sagittal) and DWI/ADC maps (axial, upper abdomen, b-values: 0, 800, 1400, 2000) For contrast enhancement, DCE axial plane images and fat suppression T1 axial images were taken. This was added to the revised manuscript.
2) Specific comments for revision
1. p.1 Abstract
In your abstract, there is no definition of a low PI-RADS score. You should mention in the abstract. For example, the low PI-RADS (<=2).
Thank you for this comment. We have added the definition of high and low PI-RADS scores in the abstract.
Round 2
Reviewer 2 Report
I confirmed that the correction was made according to the comment.